# Emergence of a New Strain of DENV-2 in South America: Introduction of the Cosmopolitan Genotype through the Brazilian-Peruvian Border

**DOI:** 10.3390/tropicalmed8060325

**Published:** 2023-06-17

**Authors:** Murilo Tavares Amorim, Leonardo H. Almeida Hernández, Felipe Gomes Naveca, Ivy Tsuya Essashika Prazeres, Ana Lucia Monteiro Wanzeller, Eliana Vieira Pinto da Silva, Livia M. Neves Casseb, Fábio Silva da Silva, Sandro Patroca da Silva, Bruno Tardelli Diniz Nunes, Ana Cecília Ribeiro Cruz

**Affiliations:** 1Institute of Biological Sciences, Federal University of Pará, Belém 66075-110, Brazil; murilotavares35@gmail.com; 2Department of Arbovirology and Hemorrhagic Fevers, Evandro Chagas Institute, Health and Environment Surveillance Secretariat, Ministry of Health, Ananindeua 67030-000, Brazil; 3Oswaldo Cruz Foundation, Rio de Janeiro 21040-900, Brazil

**Keywords:** arboviruses, dengue virus serotype 2, cosmopolitan genotype

## Abstract

Dengue virus 2 (DENV-2) seriously contributes to dengue-related mortality. It includes five nonsylvatic genotypes, with cosmopolitan being the most widespread with a significant contribution to the total number of DENV-2 cases globally. In South America, the cosmopolitan genotype was first recorded in 2019 in Madre de Dios, Peru, and then in Goiás (Midwest Brazil) in November 2021. In this study, we tested 163 human serum samples from Acre (Northern Brazil) collected during a DENV outbreak between 2020 and 2021 for all DENV genotypes by RT-qPCR. Of the 163 samples, 139 were positive for DENV-2, and 5 were positive for DENV-1. Five DENV-2-positive samples from early 2021 were sequenced, and the sequences clustered with the three other DENV-2 cosmopolitan genotype sequences already recorded on the continent. These results create a geographical link, suggesting the possible route of introduction of the DENV-2 cosmopolitan genotype into Brazil through the border with Peru, from which it may have dispersed to Midwest Brazil.

## 1. Introduction

Dengue virus (DENV) is a major cause of morbidity and mortality worldwide, leading to recurrent epidemics in urban and rural areas. DENV includes four antigenically distinct and genetically similar serotypes, which show approximately 65% genome identity [1]. Infections caused by DENV can be either subclinical or cause a range of diseases, ranging from a mild flu-like syndrome with a rash, known as dengue fever (DF), to a severe and sometimes fatal illness characterized by capillary leakage, thrombocytopaenia, and occasionally hypovolemic shock (DHF). Phylogenetic studies have demonstrated the presence of distinct DENV-2 genotypes associated with the occurrence of DHF [2,3].

The most recent epidemiological reports on epidemics in South America have described a wide circulation of serotypes DENV-1 and DENV-2, with DENV-2 seriously contributing to dengue-related mortality [4]. In Brazil, dengue viruses circulate in all regions, causing outbreaks and epidemics alternately among Brazilian states [5]. According to Brazilian Ministry of Health reports, 2,956,149 suspected cases of dengue fever were registered from 2020 to 2022, with 1798 deaths. Cocirculation of serotypes DENV-1 and DENV-2 was also observed at a high frequency, as in other South American countries [4,5,6,7,8,9]. In this period, the outbreaks occurred simultaneously with the SARS-CoV-2 pandemic, contributing to less detailed investigation and consequent underreporting of epidemiological surveillance data in several Brazilian states.

DENV-2 includes five nonsylvatic genotypes: American–Genotype I (restricted to Central and South America and already extinct); Asian I and Asian II–Genotypes IV and V (Asian continent countries); Asian/American–Genotype III (Central and South America, and Southeast Asia); and Cosmopolitan–Genotype II (Asia-Pacific, Middle East, and Africa). The last is the most widespread, with a significant contribution to the total number of DENV-2 cases globally [10,11].

Much has been discussed about genomic diversity, evolution, and transmission dynamics in regions favourable to virus dissemination. The high geographic circulation creates potential conditions for the proliferation of viral agents and their genotypes and lineages. This determinant, added to the high growth rate of spatial distribution, sets favourable conditions for the dissemination of vectors, modifications of evolutionary dynamics, and remodelling of enzootic amplification of DENV [12].

Since the introduction of DENV-2 in Brazil in the 1990s to 2020, only genotype III was detected in the country [5,13]. A new genotype, cosmopolitan (genotype II), was detected for the first time in Brazil from a human case reported in the state of Goiás (Midwest region) in November 2021 [14]; this was the second official record of this genotype in the Americas following an outbreak in Peru in 2019 [11]. The increased dengue severity on the continent may be related to the shift from single-serotype endemicity to hyperendemicity owing to the cocirculation of multiple serotypes and the introduction of new genotypes [3,11,15].

Based on an understanding of the evolutionary dynamics and global dispersion of DENV-2 in the Americas, we can presume that there has been more expressive heterogeneity of the cosmopolitan genotype compared to the other DENV-2 genotypes. Despite the phylogenetic relationship between the single case of the cosmopolitan genotype in the Midwest region of Brazil and the cases described in Madre de Dios, Peru [11,14], no geographic link between these regions, which do not share borders, has been established [14,16,17].

Here, we describe cases of the cosmopolitan genotype in Brazil reported during a DENV-2 outbreak in the state of Acre (Northern region) in early 2021. These cases create a closer geographic link between the findings from Peru and the Midwest region of Brazil, clarifying the most likely route of introduction of this genotype to Brazil from the border with Peru at the Madre de Dios region.

## 2. Materials and Methods

### 2.1. Study Samples

The study samples were selected within an epidemiological context in the period 2020–2021 during an outbreak in the state of Acre, northern Brazil. A total of 163 serum samples were obtained from humans clinically suspected of having dengue fever. The samples were tested and confirmed for DENV-2. Subsequently, cells were isolated in cell culture.

### 2.2. RNA Extraction

For RNA extraction, 140 µL of cell culture supernatant was used using QIAamp^®^ Viral RNA Mini Kit (Qiagen, Hilden, Germany) following the manufacturer’s recommendations. After purification, the RNA was quantified using a Qubit RNA HS Assay Kit (Thermo Fisher Scientific, Waltham, MA, USA) with Qubit 3.0 equipment (Thermo Fisher Scientific) according to the manufacturer’s recommendations. The samples were stored at −70 °C (under a range of −60 °C to −80 °C) until use.

### 2.3. Molecular Detection

The assay was performed using SuperScript III Platinum One-Step Quantitative RT-PCR Kit (Thermo Fisher Scientific) and included a set of oligonucleotide primers and dual-labelled fluorescent (Taqman) probes for in vitro, qualitative detection of DENV-1–4. Target amplification was recorded as an increase and accumulation of fluorescence over time in contrast to the background signal. The 25 µL reaction was composed of 12.5 µL of a 2× SuperScript III Platinum One-Step RT-PCR Master Mix, 2.2 µL of nuclease-free water, 1.0 µL of forward and reverse primers for DENV-1 (F–5′, CAAAAGGAAGTCGYGCAATA, 3′; R–5′, CTGAGTGAATTCTCTCTGCTRAAC, 3′), 0.5 µL for DENV-2 (F–5′, CAGGCTATGGCACYGTCACGAT, 3′; R-5′, CCATYTGCAGCARCACCATCTC, 3′), 1,0 for DENV-3 (F–5′, GGACTRGACACACGCACCCA, 3′; R-5′, CATGTCTCTACCTTCTCGACTTGYCT, 3′), 0,5 for DENV-4 (F–5′, TTGTCCTAATGATGCTRGTCG, 3′; R-5′, TCCACCYGAGACTCCTTCCA, 3′), 0.25 µL of DENV-1-DENV-4 probes, and 5 µL of extracted RNA [18].

Using the 7500 Fast Real-Time PCR system (Thermo Fisher Scientific), the RT-qPCR assays were performed under the following cycling conditions: an initial RT step at 50 °C for 30 min, a denaturation step at 95 °C for 2 min, 45 cycles of 15 s at 95 °C, and a final extension step of 1 min at 60 °C. Each sample was analysed in duplicate and considered positive when the average cycle threshold (Ct) value was less than 37. In total, 163 samples were tested. Positive samples for DENV-2 with Ct values less than 20 were selected for viral isolation and subsequent genomic analysis of the supernatants. The assay was validated by positive (DENV-1–4 lyophilized antigens) and negative (nuclease-free water) controls.

### 2.4. Virus Isolation

To obtain isolates, we used an *Aedes albopictus*-derived cell line, clone C6/36 (ATCC: CRL1 660), which was seeded in 10 mL culture tubes containing 1.5 mL of glutamine-modified Leibowitz medium (L-15) plus 2.95% tryptose phosphate, nonessential amino acids, antibiotics (penicillin and streptomycin), and 2% foetal bovine serum. The cells were inoculated in a 1:10 ratio of serum in glutamine-modified Leibowitz medium (L-15) and were observed daily for a period of 7 days or until the cytopathogenic effect (CPE) was verified. Upon completion, the infected cells were used for genomic sequencing.

### 2.5. Sequencing

Genomic analysis using next-generation sequencing (NGS) was performed on five DENV-2 isolates from human serum samples screened by RT-qPCR. The samples were prepared for sequencing by synthesizing first and second strands of complementary DNA, which were obtained with cDNA Synthesis System Kit (Roche Diagnostics, Basel, Switzerland) and 400 µM Roche random primer. Agencourt AMPure XP Reagent Kit (Beckman Coulter, Brea, CA, USA) magnetic beads were used for cDNA purification, and library preparation was performed by amplicon using the DNAprep kit (Illumina, San Diego, CA, USA) adapted to the DENV-2 oligonucleotide set. Quantification of cDNA was assessed using a Qubit 3.0 Fluorometer (Thermo Fisher Scientific), and the fragment size range was evaluated using a 2100 Bioanalyzer Instrument (Agilent Technologies, Santa Clara, CA, USA). Sequencing was performed using the MiniSeq sequencing system (Illumina) [19].

### 2.6. Bioinformatic Analysis

The generated reads were initially mapped against the DENV-2 reference genome (NC_001474) available in the National Center for Biotechnology Information (NCBI) database to obtain consensus sequences using Geneious v.9.1.8 [20] software. The coding regions of five obtained sequences (two complete and three partial) were aligned to 124 other sequences of different genotypes of DENV-2 available in GenBank using MAFFT v.7 [21] software. Then, a maximum likelihood (ML) phylogenetic inference was built [22] with 1000 bootstrap iterations using the IQ-TREE v.1.6.12 program [23]. GTR+F+I+G4 was selected as the best nucleotide substitution model, according to ModelFinder embedded in IQ-TREE [24]. The ML tree was rooted on the midpoint with FigTree v.1.4.4 software (http://tree.bio.ed.ac.uk/software/figtree/ (accessed on 24 February 2023)) and edited with Inkscape v.1.1 (https://inkscape.org/release/inkscape-1.1/ (accessed on 24 February 2023)).

## 3. Results

We analysed 163 serum samples from individuals clinically suspected of being infected with all four dengue serotypes using multiplex RT-qPCR and detected DENV-2 in 144 (88%), of which 139 were positive for DENV-2 and 5 for DENV-1. The results showed an increase in the number of cases for DENV-2 between January and March 2021. Of the 139 DENV-2-positive samples, five presented Ct values less than 20 and were inoculated into C6/36 cells, from which we obtained five isolates.

The five isolates were sequenced and clustered in a monophyletic clade, including the sequence from the state of Goiás, Brazil, and two Peruvian sequences. Furthermore, these genomes belong to lineage 5 of the DENV-2 cosmopolitan genotype (Figure 1). The sequences generated in this study were deposited in GenBank under the following accession numbers: OQ511271 to OQ511275. All sequences included in the phylogenetic inference are listed in Appendix A.

## 4. Discussion

The first report of a DENV-2 cosmopolitan strain described in the Americas occurred in the late 1990s in the Yucatan Peninsula of Mexico, but only one viral isolate was studied [11]. This genotype is circulating in other regions of the world and has been replacing the genotypes predominantly circulating in several regions of Asia [25]. We can infer from recent reports that a large-scale introduction of the cosmopolitan genotype of DENV-2 from Asian regions to Madre de Dios, Peru, probably occurred. We can hypothesize that the strain derives from the strains reported in Dhaka, Bangladesh, but further study is needed with regard to where its introduction occurred and how long it has been circulating. The cosmopolitan genotype has recently been reported in Madre de Dios and is circulating worldwide, presenting a wide dispersal pattern of its lineages (Figure 2) [11,14].

Until 2020, the Asian-American genotype was the only DENV-2 genotype circulating in Brazilian territory [5]. The first evidence of the circulation of the cosmopolitan genotype was obtained from a patient from Goiás in November 2021 [14]. The five sequences obtained in this study cluster into a South American clade, well supported by high bootstrap values, with the two Peruvian sequences and the only previously described sequence from Brazil. The data presented herein represent the first evidence of detection of the DENV-2 cosmopolitan genotype in the northern region of Brazil and, more specifically, in Acre months before detection in Goiás. Overall, the circulation of this novel genotype in Brazil may have implications for its attenuation, virulence, and epidemic capacity. Research has indicated genetic variations between two DENV-2 genotypes concerning clinical manifestations in humans: the Southeast Asian genotype associated with both DF and DHF and the American genotype associated with DF alone [2]. Consequently, additional investigations are necessary to explore specific nucleotide and/or amino acid changes, which will provide a more comprehensive understanding of the viral population within the host and its potential association with pathogenesis.

There are recent reports on the circulation of the new genotype based on viral isolates, and retrospectively, there are limitations to having detailed clinical information [11,14]. Therefore, to determine the route of dispersal from the point of origin, more point collection and sequencing information between Bangladesh and the Americas is needed. Consequently, a larger volume of data is needed to describe the evolutionary route.

Human serum samples were collected in February and March 2021, implying circulation of this genotype in Brazil before the case reported in Goiás [14] and two years after the outbreak in Madre de Dios, Peru [11]. The phylogenetic inference also revealed that the South American clade is positioned among sequences belonging to lineage 5 of the genotype [10], also described in the literature as Lineage C [26] or clade A [27]. These findings indicate a possible dispersal route of the DENV-2 cosmopolitan genotype in South America from Peru, with a feasible introduction into Brazil through the border of Acre and Madre de Dios, from which it may have spread to the Brazilian Midwest.

## 5. Conclusions

This study contributes to filling in gaps about the dispersion of the cosmopolitan genotype from Peru [11] into Brazilian territory, highlighting the importance of strengthening genomic surveillance of DENV-2 and conducting retrospective studies in Brazil and South America to monitor the genotype dispersion and the precise time of its introduction into the continent and Brazil.

## Figures and Tables

**Figure 1 tropicalmed-08-00325-f001:**
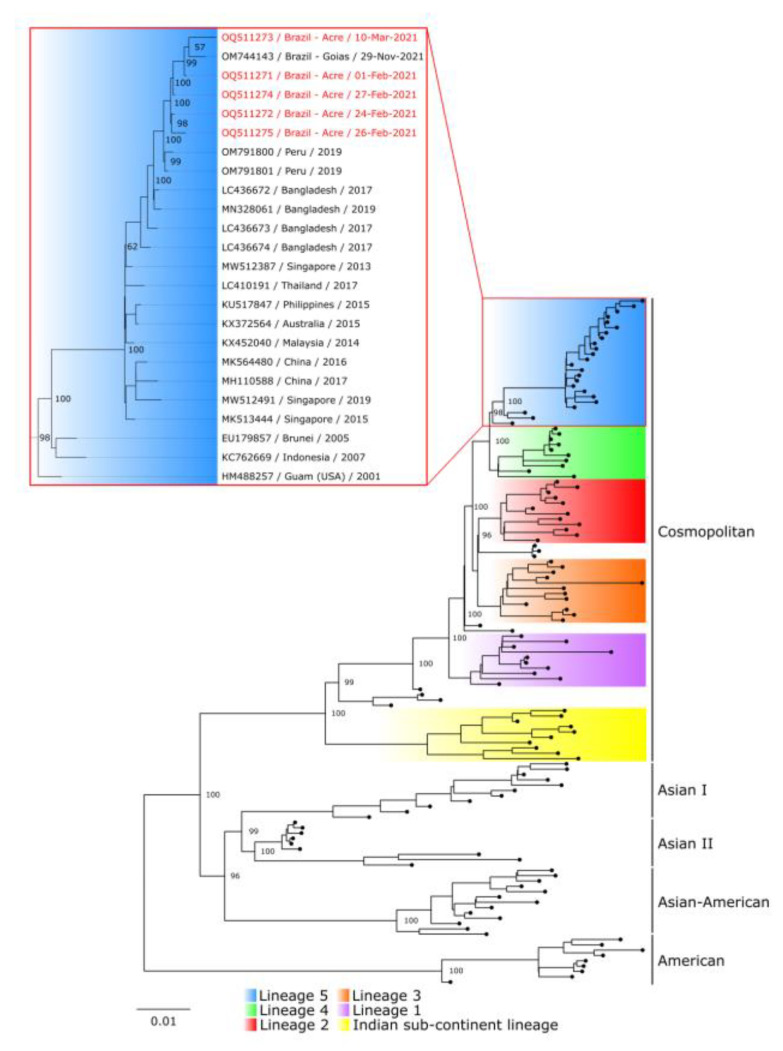
Maximum likelihood phylogenetic tree based on alignment of the coding region of a DENV-2 dataset. The five sequences presented in the study (highlighted in red) and 124 others belonging to the five nonsylvatic genotypes of the virus. The nucleotide substitution model selected was GTR+F+I+G4. The genotypes are identified on the right side of the tree, and a different colour represents each lineage of the cosmopolitan genotype. Each branch of the highlighted lineage 5 clade contains the accession number, country, and collection date of the sequence.

**Figure 2 tropicalmed-08-00325-f002:**
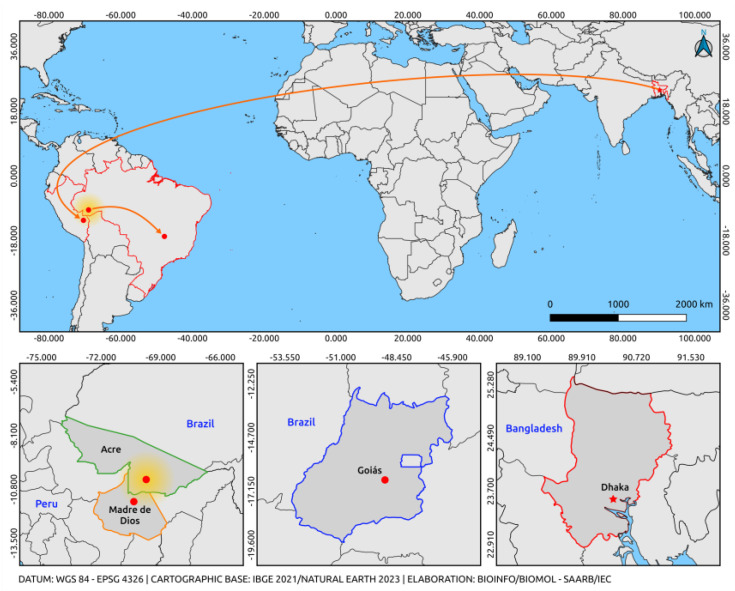
Map illustrating the likely route of introduction of DENV-2 suggested by this study. Orange arrows indicate only the exit from the point of origin and arrival by an unmapped route (Dakha/Bangladesh—Acre/Brazil), (Acre/Brazil—Goiás/Brazil), based on reports from the last five years. The colours delineate the states and the intersections between the regions of the country. The yellow gradient circle indicates the occurrence of possible dispersal of the strain to other regions of Brazil and bordering countries. The map was built using QGIS v.3.28 software, available at https://qgis.org/pt_BR/site/ (accessed on 23 April 2023).

## Data Availability

The sequences generated in this study were deposited in GenBank under the following accession numbers: OQ511271 to OQ511275.

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
