# Peer review of "Emergence of a New Strain of DENV-2 in South America: Introduction of the Cosmopolitan Genotype through the Brazilian-Peruvian Border"

_tropicalmed, 2023, doi:10.3390/tropicalmed8060325_

Round 1

Reviewer 1 Report

The paper entitled "Emergence of a new strain of DENV-2 in South America: introduction of the cosmopolitan genotype through the Brazilian-Peruvian border" is interesting and would be useful dengue response in Brazil and the region.

I have the following comments and suggestions:

  1. It is the general understanding that the South Asian DENV-2 strain is linked to severe dengue as compared with South American strains. What are the molecular differences between mild and severe dengue due to the DENV-2 virus isolated in the current outbreak? It would be to know the region of deferences in the gene.
  2. I wonder situation if multiple serotypes have been isolated from the same patient or if they were coinfected with COVID-19. It would be nice if the author could explain, which might add value to understanding the severity.
  3. What was the immunological response (primary or secondary infection) among the five strains used with a CT value of 20?
  4. What were the clinical features, frequency, and days of hospitalization that could be linked with genetic replacement?
  5. I think it is difficult to generalize the result with the limited number of cases sequenced in the study. I suggest rewriting the statement or including more cases presented with mild and severe symptoms if it is possible.

Minor typographical editing is suggested. 

Author Response

Dear,

First of all I would like to thank the evaluator for his contributions, and I attach the answer to the questions.

Reviewer 2 Report

Tavares Amorim et al., studied the introduction route of Dengue virus 2 genotype II (cosmopolitan) to central part of Brazil. By a PCR study of 163 human sera from Nothern Brazil 139 DENV-2 and five DENV-1 proved to be positives. From the DENV-2 positives with low Ct values 5 virus strains were isolated. By sequencing these isolates, and by phylogenetic studies the authors proved, that this genotype got to western part of Brazil (Acre) from a Peruvian epidemic in 2019 and later spread to central part of the country (Goias). The applied methods are correct, conclusions are supported by the results, the English usage is OK, figures and maps help in better undertanding.

spelling:

lines 113 and  167  Virus isolation

Do the authors have any proofs, or presumptions how the virus could get to Peru from Bangladesh? Did anyone studied, proved the geograhical route of this virus introduction to South America? If yes, please include into the paper.

Quality English usage is good (except vral isolation which is virus isolation).

Author Response

(The authors gave the same response as above.)

Round 2

Reviewer 1 Report

Thank you very much for responding to my concern. I have no further comments at this point in time.

Minor editing is suggested.